# Bayesian hierarchical models for disease mapping applied to contagious pathologies

**Sylvain Coly[1,2], Myriam Garrido** [1]*, **David Abrial[1], Anne-Françoise Yao[2]**

**1** Centre INRA Auvergne Rhône-Alpes, Unité d'Épidémiologie des Maladies Animales et Zoonotiques, Saint Genès Champanelle, France, **2** Laboratoire de Mathématiques UMR 6620, CNRS, Université Clermon-Auvergne, Aubière Cedex, France

* myriam.garrido@inra.fr

**Data Availability Statement:** All relevant information to simulate similar simulated data sets are within the manuscript and its Supporting information files. Real Data are property of the French "Direction de l'alimentation"

## Abstract

Disease mapping aims to determine the underlying disease risk from scattered epidemiological data and to represent it on a smoothed colored map. This methodology is based on Bayesian inference and is classically dedicated to non-infectious diseases whose incidence is low and whose cases distribution is spatially (and eventually temporally) structured. Over the last decades, disease mapping has received many major improvements to extend its scope of application: integrating the temporal dimension, dealing with missing data, taking into account various a prioris (environmental and population covariates, assumptions concerning the repartition and the evolution of the risk), dealing with overdispersion, etc. We aim to adapt this approach to model rare infectious diseases proposing specific and generic variants of this methodology. In the context of a contagious disease, the outcome of a primary case can in addition generate secondary occurrences of the pathology in a close spatial and temporal neighborhood; this can result in local overdispersion and in higher spatial and temporal dependencies due to direct and/or indirect transmission. In consequence, we test models including a Negative Binomial distribution (instead of the usual Poisson distribution) to deal with local overdispersion. We also use a specific spatio-temporal link in order to better model the stronger spatial and temporal dependencies due to the transmission of the disease. We have proposed and tested 60 Bayesian hierarchical models on 400 simulated datasets and bovine tuberculosis real data. This analysis shows the relevance of the CAR (Conditional AutoRegressive) processes to deal with the structure of the risk. We can also conclude that the negative binomial models outperform the Poisson models with a Gaussian noise to handle overdispersion. In addition our study provided relevant maps which are congruent with the real risk (simulated data) and with the knowledge concerning bovine tuberculosis (real data).

## Introduction

Disease mapping aims to determine the underlying disease risk from scattered data [1] and to represent it on a smoothed colored map [2]. This methodology has been introduced by Besag [3] to study the distribution of different cancers in the USA [4]. Since then, its scope of

(https://agriculture.gouv.fr/mots-cles/dgal%20-%20bsa.sdspr.dgal@agriculture.gouv.fr@) and cannot be provided by the authors. Moreover legal restrictions exist in France for this kind of data: when a disease case occurs in a geographical unit with few population (as observed in our dataset), anonymized data does not guaranty that the identity of the case cannot be discovered. Thus the real dataset cannot be provided.

**Funding:** This work has been realized thanks to the financial support of INRA (Institut Nationnal de la Recherche Agronomique, France, http://www.inra.fr/) and the Auvergne region (France, now région Auvregne Rhône-Alpes, https://www.auvergnerhonealpes.fr/) who co-funded the Ph.D. of S. Coly. The funders had no role in study design, data collection and analysis, decision to publish, or preparation of the manuscript.

**Competing interests:** The authors have declared that no competing interests exist.

application has significantly expanded: it is used for a wide range of human and animal pathologies and also in many other fields of application (climatology, prevention of natural disasters, etc). However, this methodology keeps being dedicated to non-infectious diseases whose incidence is low and whose distribution is spatially (and eventually temporally) structured. The distribution of cases must be quite similar to a Poisson distribution, thus the incidence of the studied phenomenon should be low. As disease mapping is a spatiotemporal smoothing method, naturally data must be spatially and temporally dependant. As this approach is applied on non-contagious pathologies, the spatial and temporal correlations are classically due to the structure of unknown environmental cofactors. They may also be linked to the repartition and structure of the population.

Disease mapping methodology is based on Bayesian inference, which aims to estimate parameters using the realization of events and some assumptions concerning the studied phenomenon. These assumptions can include some covariates and processes which handle the structure of the risk, according to the specific knowledge of the epidemiologists, in order to fit the model to the problem. The Bayesian framework particularly suits disease mapping issues as it models limited data; indeed for each period and each area, a single value is recorded. Since its introduction, the disease mapping methodology received many major improvements to adapt to other contexts: modeling various rare non-infectious diseases [5, 6], taking into account environmental and population covariates [7–10], integrating the temporal dimension [11, 12], dealing with missing data [13–15] or overdispersion [16–18], etc.

An important development would be to apply this approach to rare infectious pathologies. If for frequent diseases, representing the cases over the population provides very informative maps, it is not the case for rare pathologies. In fact, the few cases bring very limited information. For non-infectious phenomena, disease mapping methodology provides relevant risk maps handling the spatial (or spatiotemporal) correlation of the data implied by factors related to the environment or the population. In the context of a contagious disease, the outcome of a primary case can in addition generate secondary occurrences of the pathology in a close spatial and temporal neighborhood. In this case, besides the correlations due to external factors, extra-dependencies are due to contagion (direct transmission) and/or indirect transmission (for diseases vectorized for instance by ticks, flies, mosquitoes, etc). The outcome of a primary case highly increases the probability of the outcome of secondary cases in a close spatial and temporal neighborhood. Depending on the spatial and temporal scales, the secondary cases may occur in the same area and during the same period, thus local overdispersion may appear, due to the high frequence of null values and the occurrence of extreme values. The secondary cases may also occur in the neighboring regions and/or during the following periods, thus infectious diseases generally show higher spatial and temporal dependencies.

Other statistical methods are dedicated to the analysis of contagious data. On the one hand, when the exposed individuals are clearly identified and when the relations between them are well known, graphs and networks provide a relevant framework to study the spread of contagious diseases. When the literature fully describes an ecological system including a pathogen and its hosts, dynamic models are very relevant. However both of these methods cannot be applied at a much larger scale (e.g., country), because the links between all individuals and all the mechanisms explaining the propagation cannot be obtained. On the other hand, monitoring methods of outbreaks can be used to study epidemic diseases as for instance influenza or acute gastroenteritis. Nevertheless this approach is only relevant when the incidence of a pathology is high enough. As a conclusion, there is a lack of statistical methods to evaluate the risk at the scale of a country in the context of contagious diseases whose incidence is low.

The aim of our study is to propose a generic disease mapping method dedicated to rare infectious diseases, i.e., diseases for which the outcome of a case involves the likely occurrence

of other cases in their (spatial and temporal) neighborhood and for which the number of cases remains quite low. In this context, data are generally locally overdispersed and strongly spatially and temporally structured. We test flexible distributions at the first level to handle local overdispersion. We also consider lots of deterministic functions and random processes to model the spatial (and enventually temporal) dependencies. Relevant methodological choices would provide efficient disease mapping models to describe data concerning rare infectious pathologies. Thus we test a wide range of models in order to determine which ones are the most adapted to handle contagion. We first recall the structure of the Bayesian hierarchical models used in the context of disease mapping and explains our methodological choices. Then we describe the contagious framework of our study, in particular our bovine tuberculosis data and the way we simulated other datasets, congruent with an infectious pathology. In our results we show the best models to fit contagious data and highlight the interest of such an approach in spatial epidemiology. We conclude giving a short synthesis concerning our results and points out some relevant prospects.

## Materials and methods

### Bayesian hierarchical models for disease mapping

Three-level Bayesian hierarchical models are usually used in the context of disease mapping. The first level defines the probability distribution which rules the outcome of the cases. Its parameters depend on the size and the structure of the population [15, 19, 20] and on the relative risk in each area and for each period. The second level defines this risk and appears as a combination of environmental cofactors [7–10], of processes which handle the spatial correlations, and of an unstructured heterogeneity of data. In this context, the spatial (and eventually temporal) dependencies can be considered as the expression of unknown covariates. The third level defines the prior distributions of all the parameters introduced in the first two levels. Finally, we deal with model selection, since many models are tested.

Since the first works on this topic, the components of the three-level Bayesian hierarchical models have considerably evolved to address different issues: inclusion of environmental and population covariates, integration of the temporal dimension, adaptation to missing or overdispersed data. More precisely, various distributions are used at the first level to describe the outcome of the phenomenon of interest, and lots of deterministic functions and stochastic functions have been tested at the second level to describe the structure of the risk. We aim to apply disease mapping to contagious pathologies. In the general context of infectious diseases, the secondary cases can occur in the same area and during the same period, or in a close spatial and temporal neighborhood of the primary case. Other specific transmission modes (influenced for instance by the behavior of the individuals and/or their movements) can also have an important influence on dissemination. But they are disease specific and would not be considered in this article as we want to study a generic methodology.

The contagion can imply both local overdispersion (when secondary cases are in the same geographical unit at the same time) and a strong spatial and temporal structuration (when secondary cases are in a close spatial and/or temporal neighborhood). Therefore the models considered to handle contagion must take into account both aspects. On the one hand, the local overdispersion can be handled by overdispersed distributions (*in extenso* distributions which allow values of mean and variance to be different) and/or by random processes which model the individual heterogeneity. On the other hand, high spatial (and enventually temporal) correlations are handled by functions (of position and/or time) and/or structured random processes at the second level at the Bayesian hierarchical models. In the following sections we aim

to present the main modelization possibilities and to highlight which methodological choices seem relevant to address infectious diseases.

Another specificity of our study is that we do not focus on a specific disease, even if we illustrate this study with bovine tuberculosis data. In fact we neither try to explain the causes nor to define the risk factors of this disease, thus we do not consider cofactors in our models. More precisely we consider no stratification of the population by covariates at the first level of the hierarchical models, and no environmental covariates at the second level. However, if one focuses on a given precise phenomenon, one can easily add relevant cofactors, as in most of the studies in the literature.

In that follows, we consider a studied region $\mathcal{A}$ divided into $n$ geographical areas $\{A_i\}_{i \in [[1,n]]}$. The duration of study is divided into $m$ periods $\{\tau_j\}_{j \in [[1,m]]}$. We assume that $Y = (Y_{ij})_{ij}$ is a set such that $Y_{ij}$ is the number of cases in $A_i$ ($i \in [[1, n]]$) for the period $\tau_j$ ($j \in [[1, m]]$). Most of the analyses in the literature consider data aggregated over administrative areas, even if they are very irregular. Indeed these have very different shapes and sizes, and they are obviously non-equivalent. In this study, we consider spatial data which are aggregated in small hexagons in order to avoid this potential bias.

**Probability distribution of cases (first level).** The first level sets the distribution $\mathcal{D}ist$ of the number of observed cases $Y_{ij}$ for any $(i, j) \in [[1, n]] \times [[1, m]]$,

$$Y_{ij} \sim \mathcal{D}ist.$$

The mean of this distribution is

$$\mathbb{E}(\mathcal{D}ist) = E_{ij}.R_{ij}$$

where $E_{ij}$ is the expected number of cases and $R_{ij}$ is the relative risk in the area $A_i$ during the period $\tau_j$. $E_{ij}$ is usually calculated under the assumption of a constant disease rate in $\mathcal{A}$, so this value is proportional to the size of the population in $A_i$, which is supposed to be constant during the time period.

The Poisson and binomial distributions have been firstly used at the first level in the context of disease mapping. As the population size is large and the incidence of the studied pathologies is low, the Poisson distribution is particularly suitable to model such data. Even if the binomial distribution was employed in several early disease mapping studies, most of the spatiotemporal studies still model cases as the realization of a Poisson distribution

$$Y_{ij} \sim \mathcal{P}(\mu_{ij})$$

[11, 21, 22]. Besides this distribution being the most commonly used to model count data, it especially appears as the standard distribution in the disease mapping setting. Moreover, it involves only one parameter and less complexity. We deal with the Poisson distribution in order to compare classical models to the ones we particularly propose. Nevertheless other distributions have been tested in several studies to handle various problems coming from the data, such as for instance overdispersion, missing values, under-detection of the pathology, etc. In particular, two (or more) parameter distributions (negative binomial [16, 17], generalized Poisson [18], etc) have been tested to handle the potential overdispersion of data. This overdispersion can result in the outcome of extreme values and in the high frequence of null values. Even if the Poisson distribution is not really adapted to overdispersed data, the unstructured heterogeneity is modeled, in many studies, at the second level with a Gaussian white noise. In the context of rare infectious pathologies (we deal with), the outcome of secondary cases in the same geographic area and during the same period may increase the local overdispersion; thus, we test a two-parameters distribution, which would be more flexible and more

adapted to handle locally overdispersed data. We consider the negative binomial distribution ($\mathcal{NB}$):

$$Y_{ij} \sim \mathcal{NB}(\mu_{ij}, \mu_{ij}.s_{ij})$$

where $s_{ij}$ is the local overdispersion. The negative binomial distribution has been used in several studies [17, 23], especially for data with high variability. The $\mathcal{NB}$ model may better fit the data as the incorporation of a second parameter increases its flexibility, but it also increases the global model complexity. Zero-Inflated (ZI) distributions have been considered in order to deal with missing data [13–15]. As we do not focus on a given pathology, we do not expect missing values, thus in our context the ZI distributions do not appear as the most relevant ones. However, if under-detection of the studied disease is suspected, one can easily replace the Poisson and the negative binomial distributions with ZI distributions.

The Bayesian framework allows using the scientific knowledge concerning the disease of interest, especially concerning the links between the pathology and personal factors (sex, age, ethnic origin, etc). Indeed, these factors can be taken into account at the first level of the model [20, 24]. If the structure of the population is different in the subdivisions of the area of interest, and if a population cofactor has a significant influence on the outcome of the pathology, the population can be stratified and the resulting data are standardized by categories.

**Structure of the risk (second level).**   In the framework of disease mapping studies, the second level of Bayesian hierarchical models is dedicated to the environmental cofactors related to the pathology of interest, and to the remaining structure of the risk $R_{ij}$. More precisely, at this level, $\ln(R_{ij})$ is defined as the sum of covariates and terms which take into account the spatial and/or temporal correlations. Thus the log-linear model is defined by

$$\ln(R_{ij}) = Cov_{(ij)} + U_{ij} + T_{ij} + V_{ij} + \epsilon_{ij},$$

where $Cov_{(ij)}$ is a combination of (generally fixed) effects due to the known cofactors which can vary among both time and space, $U_{ij}$, $T_{ij}$ and $V_{ij}$ respectively describe the spatial, temporal and spatiotemporal structuration of data, and $\epsilon_{ij}$ handles the residual unstructured heterogeneity of the distribution of the cases. This log-linear combination of spatially and temporally structured terms seems relevant since, as, in our context of infectious diseases, one case involves several other ones, neighboring in time and space.

*Spatial term $U_{ij}$.* As all the population and environmental factors related to a disease cannot be easily determined, even for non-infectious diseases, significant spatial dependencies subsist in the data from neighboring regions [2]. Thus, most of the disease mapping studies focus on the determination of the most relevant spatial processes. A spatial trend has been considered for $U_{ij}$ [11], however it implies a strong *a priori* on the distribution of the risk. A classification of the region into areas showing different risk levels has also been tested [25]; this approach seems particularly relevant if one suspects a strong influence of an environmental cofactor. Many spatial processes have been introduced to model spatial correlations without making assumptions concerning the repartition of the risk. Thus various Conditional AutoRegressive processes (CAR) have been created for $U_{ij}$: the Intrisic AutoRegressive process [26] the Besag-York-Mollie model [27], Cressie's model [28] and Leroux's model [29]. These CAR models are the most popular terms to handle spatial correlations and to provide smooth maps of the risk.

*Temporal term $T_{ij}$.* Handling the temporal evolution of the geographical distribution of the cases has been one of the major improvements of the disease mapping methodology [11, 12]. Since 2000, most of the studies in the literature have integrated temporal components. Initially, deterministic functions of time have been tested for $T_{ij}$: 1) the temporal trend [11], 2) polynomial functions of time [30] and 3) splines [21]. Even if more flexible functions have then been

tested for $T_{ij}$, they impliy strong assumptions concerning the temporal evolution of the pathology. Moreover, for these deterministic functions, the disease risk is supposed to follow the same evolution in all the geographic areas. Thus, stochastic processes have also been tested. In fact, they appear relevant for contagious data as the outcome of a case highly increases the occurrence of other cases in the neighborhood, when the evolution of the risk is not similar for every areas as in the case of contagion. Temporal Random Walks (RW) [10, 31] have been used to model $T_{ij}$. Finally more and more complicated AR processes have been considered [17, 22], including temporal CAR processes [15, 31]. In this case, each period has for only neighbor the previous period. These processes seem more flexible, and thus more relevant to model random temporal evolutions and to smooth the risk over time. Nevertheless some of these processes (splines, polynomial functions, high order AR processes) depend on many parameters; it can be difficult to estimate them if the study period is short.

*Spatiotemporal term $V_{ij}$.* In earlier analyses, the spatiotemporal term consists in a linear function of time $V_{ij} = a_i.\tau_j$ whose regression coefficient $a_i \in \mathbb{R}$ (for $i \in [[1, n]]$) depends on the location [11, 21, 32]. Such a spatiotemporal interaction corresponds to a strong assumption, since it models a smooth temporal variation from the initial map; it results in quite similar risk values for neighboring regions but allows distinct temporal evolutions for each area. However, most of the studies which model the spatiotemporal interaction use stochastic processes, especially temporal versions of spatial terms for $V_{ij}$. Indeed, Nobre (2005) tested the relevance of Intrinsic AutoRegressive (IAR) processes with variance depending on the period [33]. In the same way, spatial CAR terms which vary over time have been introduced for $V_{ij}$ in some studies [7, 8, 24]. They seem able to model flexible evolutions of the risk in different regions. However they do not appear well-adapted to model a diffusion of the risk as they do not consider a real spatiotemporal neighborhood. Few publications [34–36] proposes a particularly relevant approach to model propagation as it assumes the influence of past values measured in the neighborhood for the estimation of the risk [34].

*Spatial and temporal terms to handle infectious diseases.* In the context of contagion, a case may involve several other ones, close in time and space; thus the contagion may result in a reinforcement of the spatial and/or temporal structure. For this reason, we particularly want to test the relevance of spatial, temporal and spatiotemporal processes in the structure of the risk. To avoid making strong assumptions on the spatial or temporal evolution of the studied diseases, we ignore all the deterministic functions of time or space. We consider spatial, temporal and spatiotemporal Conditional AutoRegressive (CAR) processes since they are particularly convenient to produce smoothed maps when the risk is well structured. Moreover CAR processes are the most popular approach to model spatial correlations. In addition, we ignore *a priori* how the heterogeneity of the risk is structured, thus it seems relevant to consider similar processes for each term $U_{ij}$, $T_{ij}$ and $V_{ij}$. Then we consider

$$\ln(R_{ij}) = U_{ij} + T_{ij} + V_{ij} + \epsilon_{ij}$$

and test all its sub-models, where $\epsilon_{ij}$ handles the unstructured heterogeneity and $U_{ij}$, $T_{ij}$ and $V_{ij}$ are CAR processes expressed as

$$U_{ij} \sim \mathcal{N}\left( \sum_{i' \in \delta_i} U_{i'j} / \mathrm{Card}(\delta_i), \sigma_1 \right),$$

$$T_{ij} \sim \mathcal{N}\left( \sum_{j' \in \delta_j} T_{ij'} / \mathrm{Card}(\delta_j), \sigma_2 \right),$$

$$V_{ij} \sim \mathcal{N}\left(\sum_{(i',j')\in\delta_i\times\delta_j} V_{i'j'}/(\mathrm{Card}(\delta_i).\mathrm{Card}(\delta_j)), \sigma_3\right),$$

where $\delta_i = \{i' \in [\![1,n]\!]/A_i\mathcal{R}_s A_{i'}\}$, $\delta_j = \{j' \in [\![1,m]\!]/\tau_j\mathcal{R}_t\tau_{j'}\}$ and $(\sigma_1, \sigma_2, \sigma_3) \in (\mathbb{R}_+^*)^3$. $\mathcal{R}_s$ and $\mathcal{R}_t$ are respectively defined as the spatial and temporal neighborhood relations, i.e., $A_i\mathcal{R}_s A_{i'} \Leftrightarrow A_i$ and $A_{i'}$ are adjacent, and $\tau_j\mathcal{R}_t\tau_{j'} \Leftrightarrow \tau_j$ and $\tau_{j'}$ are adjacent. For each period, we choose as the temporal neighbors both the past and the future periods. Indeed in the framework of retrospective analyses, considering both past and future values provides more information and leads to more robust estimates. Following this idea and the approach of Knorr-Held, we propose a spatiotemporal CAR process for which each value is influenced by the past and the future values in the spatial neighborhood.

*Unstructured heterogeneity $\epsilon_{ij}$.* In most of the studies, the linear predictor includes a Gaussian noise with a constant variance to take into account the individual unstructured heterogeneity [22, 31, 32]. Data related to infectious pathologies are particularly overdispersed. A Gaussian term whose variance depends on the period and the region has been tested to model particularly strong heterogeneity [34, 37], however it implies a very high number of parameters and may result in very noisy maps. Gaussian components have been also used to model structured heterogeneity by adding an aggregation effect. For instance, Abellan (2008) considered a more flexible alternative by summing two Gaussian variables whose variance is very different [38]. Such a term seems relevant when the areas or the period can be divided into two different classes. We choose the Gaussian white noise

$$\epsilon_{ij} \sim \mathcal{N}(0, \sigma_4),$$

where $\sigma_4 \in \mathbb{R}_+^*$, to model the overdispersion of the data. This term is considered in the literature as a relevant approach to model the unstructured heterogeneity. To take into account the local heterogeneity, the relevance of both the binomial negative distribution and the Gaussian noise can be compared.

*Weights.* In his studies, Cressie introduced weight coefficients [28] for the structural components in order to quantify the contribution of each structural component [37]. Such components can facilitate the model fitting but can also increase model complexity. Each of the CAR terms we consider may explain the structure of the risk. However, it can be complicated to determine the contribution of each of these components. Thus, in this context, the weight coefficients introduced by Cressie seem relevant [28]. They can also improve or worsen the convergence of simulated MCMC (Markov Chains Monte Carlo). For these reasons we want to test models in which the risk is

$$\ln(R_{ij}) = a.U_{ij} + b.T_{ij} + c.V_{ij} + \epsilon_{ij} \qquad (1)$$

together with all its sub-models, with or without all the weight coefficients, *in extenso* with $a$, $b$, $c$ equal to 0, 1 or to be estimated.

**Hyperparameters (third level).**   The third level of Bayesian hierarchical models consists in a set of conditions which determine the prior distributions of the parameters of the two first levels. In general, no assumption based on the knowledge of the phenomenon of interest concerning such parameters are assumed as they are too weakly related to this phenomenon. This is why non-informative distributions should be considered, as for instance the infinite uniform improper distribution. This third level can impact the convergence of the method, so authors use distributions and values which enable and facilitate the convergence of their models. In practice, to ease the convergence, they choose poorly informative distributions (instead of

non-informative distributions) as for example the uniform distribution $\mathcal{U}(0, M)$ where $M > 0$ is large [39], or the Gamma distribution $\Gamma(\epsilon, \epsilon)$ where $\epsilon > 0$ [17, 23, 34]. As Gaussian distributions are very commonly used to describe the structure of the risk, variance is the parameter which is the most frequently used at the third level. In most of the studies, the precision parameter (inverse of the variance) follows a Gamma distribution whose parameters have very low values. Thus, if $\tau_i = 1/\sigma_i$, where $\sigma_i$ is the variance of the IAR and Gaussian variables, we assume $\tau_i \sim \Gamma(0.01, 0.01)$ for $i \in \{1, 2, 3, 4\}$ in accordance with studies of Knorr-Held, Lowe and Charras-Garrido [17, 23, 34]. Much less numerous studies using the weight coefficients have been realized, however some consider a Gamma distribution for the weight coefficients suggested by Cressie [28], thus we assume $a, b, c \sim \Gamma(5, 5)$.

**Deviance information criterion.** Most of the studies in disease mapping use the Deviance Information Criterion (DIC), introduced by Spiegelhalter in 2002, as a model comparison method [40]. The DIC is the sum of the expectation of the deviance $D$, calculated by $D(\theta) = -2\log(\mathbb{P}(y|\theta)) + C$ (with $\theta$ the set of unknown parameters of the model, $y$ the data, $C$ a constant), and the effective number of parameters, noted $p_D$ and defined as $p_D = \bar{D}(\theta) - D(\bar{\theta}) = \mathbb{E}^\theta(D(\theta)) - D(\mathbb{E}^\theta(\theta))$ by Spiegelhalter [40]. This criterion penalizes both the non-fitness (deviance) and the complexity of the model (effective number of parameters). Models with smaller DIC values are considered to be better. This generalization of the Akaike and the Bayesian Information Criterions (AIC and BIC) is described as particularly efficient when the estimations have been obtained by MCMC simulations and follow a Gaussian distribution [41]. The DIC is used in every spatiotemporal disease mapping study [22, 24, 34] and is described as particularly relevant to compare nested models [41]. This indicator can be easily computed and is meant to be a good trade-off between goodness-of-fit and complexity. The DIC appears as the only criterion dedicated to this context, in particular if the underlying risk is not known, thus this criterion appears as the most relevant approach for our study. Some studies show that this criterion can overfit the data and some difficulties to estimate the effective number of parameters have been identified [42–45]. However the variants of the DIC which have been developed in order to handle its main drawbacks can be difficult to compute or can't be used when the underlying risk is unknown, and all of them are only used in isolated studies. Thus we choose the DIC to compare our models.

## Application of disease mapping methods to contagious data

Our models are applied on both real and simulated data. Concerning real data, we choose the bovine tuberculosis as this pathology is quite uncommon in France and is an infectious disease. The aim of this Section is to describe our bovine tuberculosis data, and show how we simulated data in order to test our methodological choices. We then give an exhaustive list of the models we tested and give some clues concerning the estimation of the parameters and their computation.

**Bovine tuberculosis data (France, 2001-2010).** We consider the Metropolitan France (excluding Corsica) as the study area. The country is divided into 448 hexagons (noted $A_i$, with $i \in [[1, 448]]$) measuring 40 kilometers. The study period extends from 2001 to 2010 and is considered year by year (noted $\tau_j$, with $j \in [[1, 10]]$). We consider cattle farms as population data. They have been provided by the DGAl (the French Directorate for Food) [46] and are supposed to be constant over time. S1 Fig shows their repartition. We notice that most of the cattle farms are situated in the Northwest, in the Center and in the far Southwest of the country. At the opposite, there are no cattle breedings in a large area around Paris and in the Southeast of France. We also remark that cattle farms are far fewer near the coasts and the borders than inland.

**Table 1. Evolution of the number of cases.**

| Year | 2001 | 2002 | 2003 | 2004 | 2005 | 2006 | 2007 | 2008 | 2009 | 2010 |
|---|---|---|---|---|---|---|---|---|---|---|
| Number of cases | 64 | 48 | 56 | 44 | 72 | 77 | 80 | 85 | 66 | 111 |
| Infected hexagons | 42 | 34 | 34 | 34 | 45 | 35 | 27 | 31 | 32 | 32 |
| Extreme values (>5) | 0 | 1 | 1 | 0 | 2 | 5 | 5 | 3 | 1 | 5 |

Bovine tuberculosis is a disease caused by a bacterium which can affect the cattle [47] and also contaminate humans [48], principally in developing countries. In France, no human case occurs and the country is officially free of the disease, *in extenso* less than 0.1% of the national herd is contaminated. Nevertheless, this pathology remains a major challenge. In fact more than 100 contaminated cattle farms are identified each year (Table 1). Tens of millions of euros are spent each year to address this issue and to prevent the outcome of other cases. Some geographical areas, in particular the Dordogne and the Côte-d'Or [49], and in a lesser extent the Camargue and the Southwest [49], are particularly concerned by bovine tuberculosis (Fig 1). Brittany and Auvergne, which are among the principal farming regions, are rather unaffected (S1 Fig) [49]. However, maintaining the officially free of TB status is a major economic and sanitary issue.

The cases are defined as contaminated cattle farms and were recorded per year and per hexagon. The location and the date of bovine tuberculosis cases were provided by the DGAl [46]. Most of the cases occur in the regions identified as at risk. Nevertheless many isolated cases occurred in other areas, in particular in the neighborhood of the Dordogne and of the Côte d'Or. The number of cases clearly increases over the time between 2001 and 2010 (Table 1). However, the number of hexagons concerned by this pathology tends to decrease during the same period. Moreover, more extreme values occur in the end of the period, especially in Côte-d'Or and in Dordogne. Thus cases tend to be more and more numerous and concentrated in the previously mentioned regions over the time.

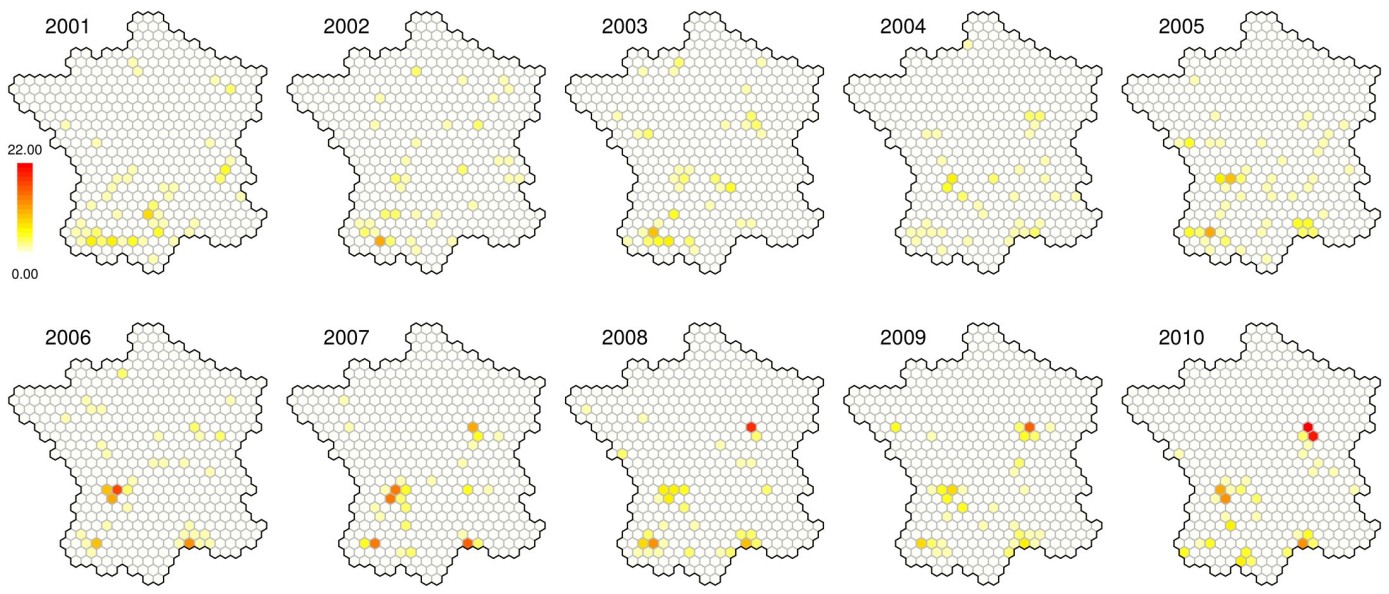

**Fig 1. Evolution of the repartition of bovine tuberculosis cases over the years.**



T1: If the area is not contaminated, the infection probability in the temporal neighborhood is 0.136.



T2: If the area is contaminated, the infection probability in the temporal neighborhood is 0.323.

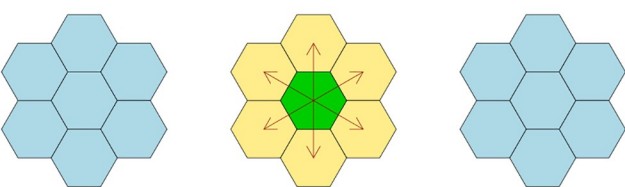

S1: If the area is not contaminated, the infection probability in the spatial neighborhood is 0.346.

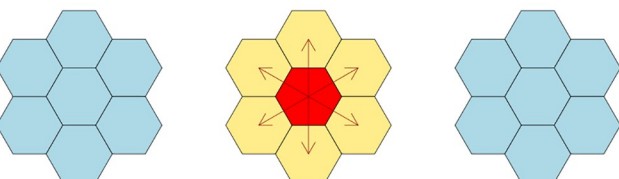

S2: If the area is contaminated, the infection probability in the spatial neighborhood is 0.635.

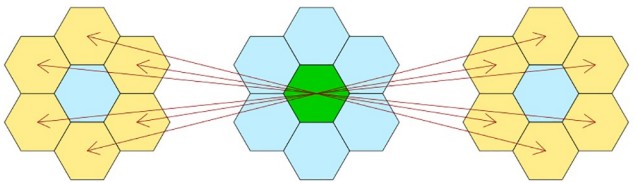

ST1: If the area is not contaminated, the infection probability in the spatiotemporal neighborhood is 0.488.

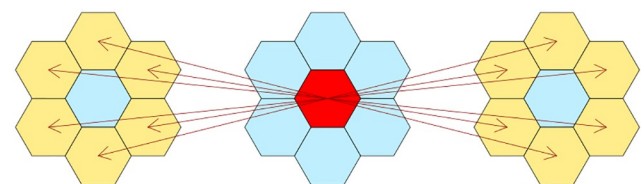

ST2: If the area is contaminated, the infection probability in the spatiotemporal neighborhood is 0.742.

**Fig 2. Correlations of areas with and without cases.**

The bovine tuberculosis data illustrates the fact that the contagion may imply high spatial and temporal correlations (Fig 2). In fact, if a hexagon is contaminated, cases occur the year before or the year after in this same hexagon (scheme T2 in Fig 2) with a probability about three times higher than if the hexagon is not contaminated (T1). Besides, if a hexagon is contaminated, cases occur at the same year in at least one adjacent hexagon (S2) with a probability which is twice the probability if there is no case in the hexagon (S1). Finally, if a hexagon is free at a date, cases occur the year before or the year after in at least one adjacent hexagon (ST1) with a probability which is two-thirds the probability if the hexagon is contaminated (ST2).

Dealing with the overdispersion from health data is an important issue in epidemiology, in particular for disease mapping. We remarked that bovine tuberculosis data (noted $Y_{ij}$, with $i \in [[1, 448]]$ and $j \in [[1, 10]]$) shows a high level of overdispersion: in fact $s_{ij} = \mathrm{Var}(Y_{ij})/\mathbb{E}(Y_{ij}) \approx 5.44$. Overdispersion may be due to spatiotemporal dependencies, however this value is particularly high and must mainly be due to the outcome of grouped cases in some areas during some periods (local overdispersion). More precisely, if we compare the distribution of cases with the quantiles of a Poisson distribution, we easily notice

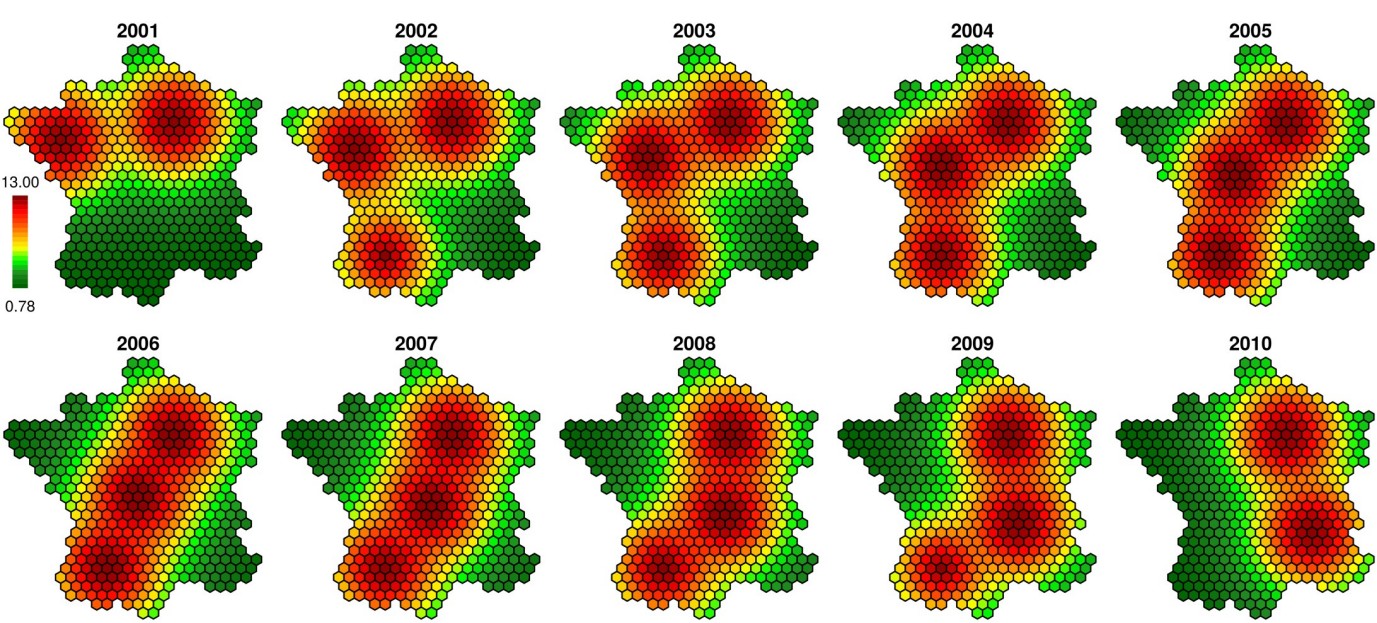

**Fig 3. Risk map with the 3 outbreaks.**

that null values are over-represented, extreme values ($> 5$) occur more frequently, and there is a lack of values equal to 1.

**Simulated data.** In addition of the analysis of real data, we test our method on simulated datasets in order to check their ability to describe the real risk for a pathology of interest, and to draw robust conclusions from our study.

*Deterministic building of the true risk.* We firstly set a uniform disease risk among the territory. In order to model basic contagious processes, we simulated in each dataset 3 circular disease outbreaks. More precisely, they were defined as 3 hotspots around which the risk exponentially decreases. The radius (around 100km) has been chosen congruently with bovine tuberculosis outbreaks which concern 3 (Dordogne) to dozens hexagons (Southwest). These 3 outbreaks (Fig 3), common to all the datasets, showed different characteristics, in order to simulate different basic scenarios:

- the first one (O1) was constant over time in risk intensity and moved from Northwest to the Southeast at a speed of about 100km per year in order to cover a large range of the country, see S2A Fig;

- the second one (O2) was constant over time in location (Northeast of the country) and risk intensity, see S2B Fig;

- the third one (O3) was constant over time in location (Southwest) while its risk intensity increased geometrically from 2001 (no outbreak) to 2005 (a maximum extend similar to O1 and O2) then decreased geometrically from 2006 to 2010, leading to a growing and then decreasing circular outbreak, see S2C Fig.

The risk values were then standardized, so their global product was 1. The outbreaks have been designed to test the detection ability of the models in 3 basic situations: stationarity, move, grow and decrease.

*Simulation of the cases*. Cases were finally simulated using population data, the global incidence of the bovine tuberculosis we used as example and resulting risk values. We distinguish 4 scenarios including or not local overdispersion (CO versus NO):

(COLR).   Constant Overdispersion and Large differences in Risk levels;

(COSR).   Constant Overdispersion and Slight differences in Risk levels;

(NOLR).   No Overdispersion and Large differences in Risk levels;

(NOSR).   No Overdispersion and Slight differences in Risk levels.

For each scenario we simulated 100 datasets using Poisson distribution when there is no overdispersion (NOLR, NOSR) and Negative Binomial distributions in case of overdispersion (COLR, COSR). This leads to a total of 400 simulated datasets. With the Negative Binomial, the overdispersion value has been chosen as $\text{Var}(Y_{ij})/\mathbb{E}(Y_{ij}) = 5.5$, in line with real data. The ratio between the lowest and highest risks is defined as 16 for Large differences in Risk levels (LR), and 4 for Slight differences in Risk levels (SR). The large differences are expected to be quite easy to detect, while the slight differences may be more challenging and explore the limits of the method. Similarly, the local overdispersion may complicate outbreak detection and the accuracy of risk estimation.

S3 Fig shows an example of a dataset obtained for the scenario (COLR). We remark that in this scenario, cases are very scattered. Congruently with the overdispersion implied by the negative binomial distribution, there is a high proportion of null values ($\approx 92\%$) and many high values occur (over 10% of the non-null values are over 5). Despite the low risk values in Brittany for the later period, many cases occur in this region. It can be explained by the very high population of cattle farms.

**Our models to handle contagion.**   As we said in Section Bayesian hierarchical models for disease mapping, we tested both the Poisson and the negative binomial distribution. At the second level, we tested all the combinations of spatial, temporal and spatiotemporal CAR processes and Gaussian white noise. We also tested the relevance of the weights suggested by Cressie to quantify the contribution of each of these processes to the global (structured and unstructured) heterogeneity of the risk values. We test 60 different models.

For the rest of this paper, the names of the different models are built with the same following structure:

- characters 1-5: distribution at the first level (“`poiss`” for the Poisson distribution and “`nebin`” for the negative Binomial distribution);

- characters 6-10: presence (“`param`”) or absence (“`nopar`”) of the weigth parameters;

- character 12: presence (“`S`”) or absence (“`x`”) of the spatial CAR process;

- character 14: presence (“`T`”) or absence (“`x`”) of the temporal CAR process;

- characters 16-17: presence (“`ST`”) or absence (“`xx`”) of the spatiotemporal CAR process;

- characters 19-22: presence (“`gaus`”) or absence (“`xxxx`”) of the Gaussian white noise.

Thus, for instance, the model named `poissnopar_S_x_ST_xxxx` is characterized by $Y_{ij} \sim \mathcal{P}(\lambda_{ij})$ and $\ln(R_{ij}) = U_{ij} + V_{ij}$ (*An exhaustive list of the tested models is given in* S1 Table).

## Estimation and computation aspects

Disease mapping methods used to require MCMC simulations to estimate the parameters. These simulations use initial data (cases and population repartition), the structure of the Bayesian hierarchical models and the eventual covariates to estimate (spatially and/or temporally) smoothed values of the risk for each region and each period. More recently Integrated Nested LAPLACE Approximations (INLA) [35, 50] are mentioned to perform the estimations. However, INLA was not of so common use at the beginning of this study. Thus, to focus comparisons with previous works on Negative Binomial distribution and spatiotemporal CAR, we chose to estimate the parameters with the software BUGS. This MCMC estimation method has been commonly used in spatiotemporal disease mapping [9, 22, 38] to perform the inference of the parameters.

The Bayesian framework with MCMC estimation implies long calculation durations in the context of intensive simulation studies and requires parallel computing to perform the analyses, thus we used the HTCondor system to allocate jobs to the different cores (about 150). We realized scripts encoded in the C++ language to treat the resulting files. We used the software R to create all the preliminary files needed by BUGS (data, scripts and models) and HTCondor (submission files), to analyze the results and to realize maps. We also used the R libraries CODA, RPostgreSQL, rgdal, spdep and scales to import the results of BUGS, to interpret the results, and to build the maps of the risk.

## Results: Best models to fit contagious simulated data

We consider a wide range of models (see S1 Table) to deal with overdispersion and chose the DIC as model comparison method. We compute the parameters for all the 400 datasets (100 replicates associated to each of the 4 scenarios) and all the 60 models. Thus these analyses provide lots of risk values, of risk maps and of DIC values whose interpretation can be delicate. Figs 4–7 show, for each model, the repartition of the DIC values based on the 100 replicates for each considered scenario. Table 2 describes the best models for each scenario according to the average value of the DIC over all the replicates. We included all the models for which the DIC does not exceed the best DIC by more than 10. We also compute average values of the DIC to determine the most relevant strategy for each scenario (Table 3).

In every analysis, models which include weight coefficients fail to fit the data. Every model with weights has a higher DIC value on average than the same model without weight coefficients. The differences between models with and without weight parameters are particularly strong in the case of overdispersed data. Besides, for each simulated scenario, the best ranking for a model with weights is always beyond the tenth place. Thus such models are always considered as irrelevant to quantify the contribution of each component to the heterogeneity of data. Thus for the rest of this section we will only consider models without weights.

We now look for the best models and the most relevant strategies to handle local overdispersion and a particularly strong spatiotemporal structure.

### Handling local overdispersion

We deal with three situations, depending on the type of dataset: overdispersed data (scenarios (COLR) and (COSR)), non-overdispersed data with a weakly contrasted risk (scenario (NOLR)) and non-overdispersed data with a weakly contrasted risk (scenario (NOSR)). The best models are quite different according to this classification.

**Overdispersed data.** The best models for the scenarios (COLR) and (COSR) have similar characteristics (Figs 4 and 5); namely all are negative binomial models without the Gaussian noise term (Table 2). These models have very similar mean values for the DIC and show the

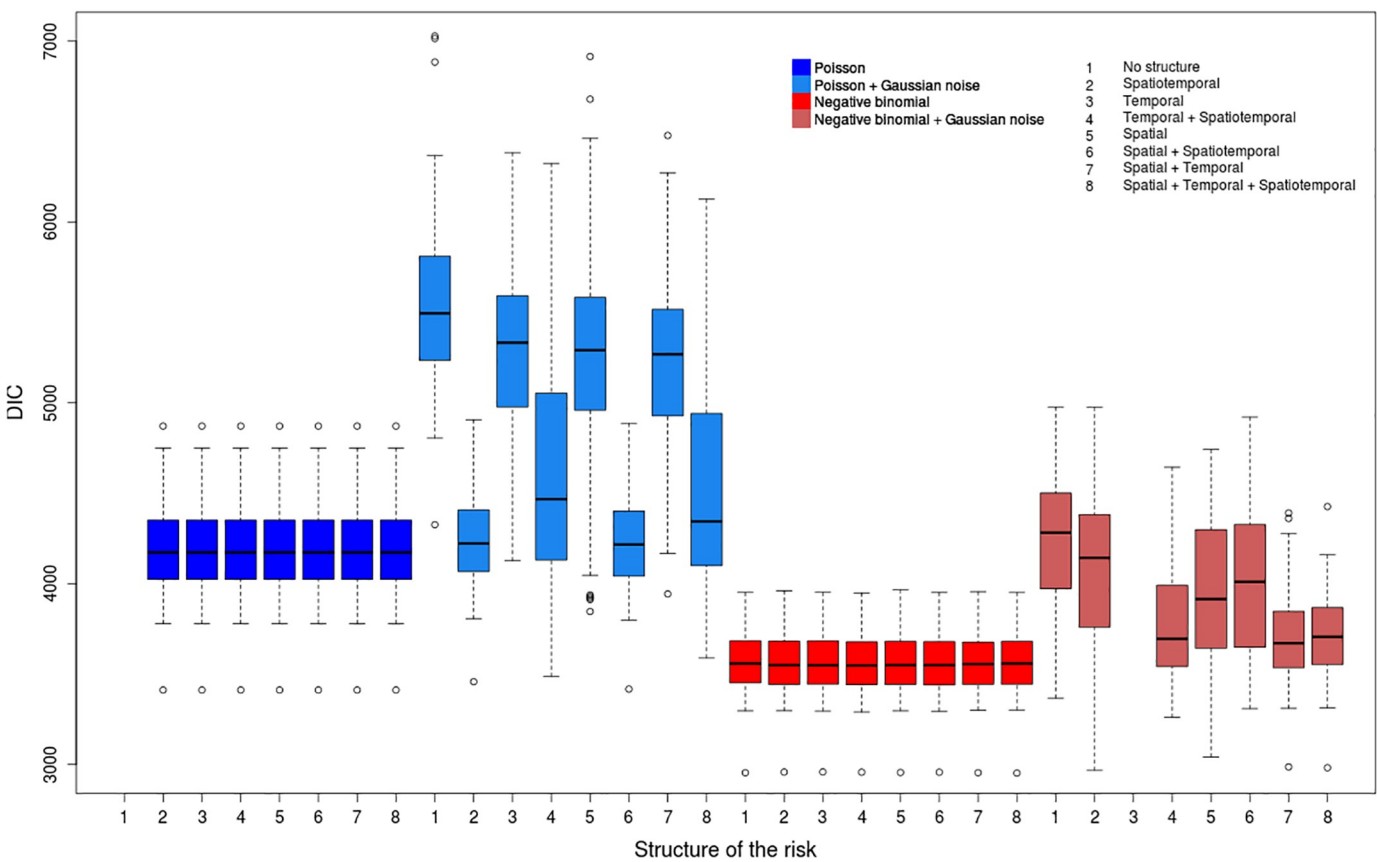

**Fig 4. Distribution of DIC values for the scenario (COLR).**

lowest DIC dispersion among all the models; however, the repartitions of the associated DIC values show slight differences (Fig 4). The Poisson models without the Gaussian noise $\epsilon_{ij}$ have also quite similar distributions despite higher DIC mean values, and they also have quite low DIC variance values. The negative binomial models with the Gaussian white noise have DIC mean values comparable to Poisson models without Gaussian noise term; however, their DIC variance values over all the replicates are particularly high, and depend also on the structure of the risk. Lastly, the Poisson models with a Gaussian noise $\epsilon_{ij}$ have DIC values with very high means and variances. In addition, for this class of models, the different structures of the risk provide very different DIC values.

One can also notice that the differences between the mean values of the DIC for the two distributions are much higher for scenarios (COLR) and (COSR) (overdispersed data, Figs 4 and 5) than for (NOLR) and (NOSR) (non-overdispersed data, Figs 6 and 7). The overdispersion particularly highlights the differences between relevant and inadequate models (Table 3).

If we consider all the 32 models without weights, we see that the negative binomial models better fit on average the outcome of cases if overdispersion is assumed (Table 3). We also notice that for each combination of the terms of the structure, the negative binomial model outperforms the Poisson model. Besides, the models with the Gaussian white noise $\epsilon_{ij}$ lead to much higher DIC values.

The results show that the negative binomial model is well adapted to model the remaining variability due to local overdispersion, even if is not tested in most of the studies in the

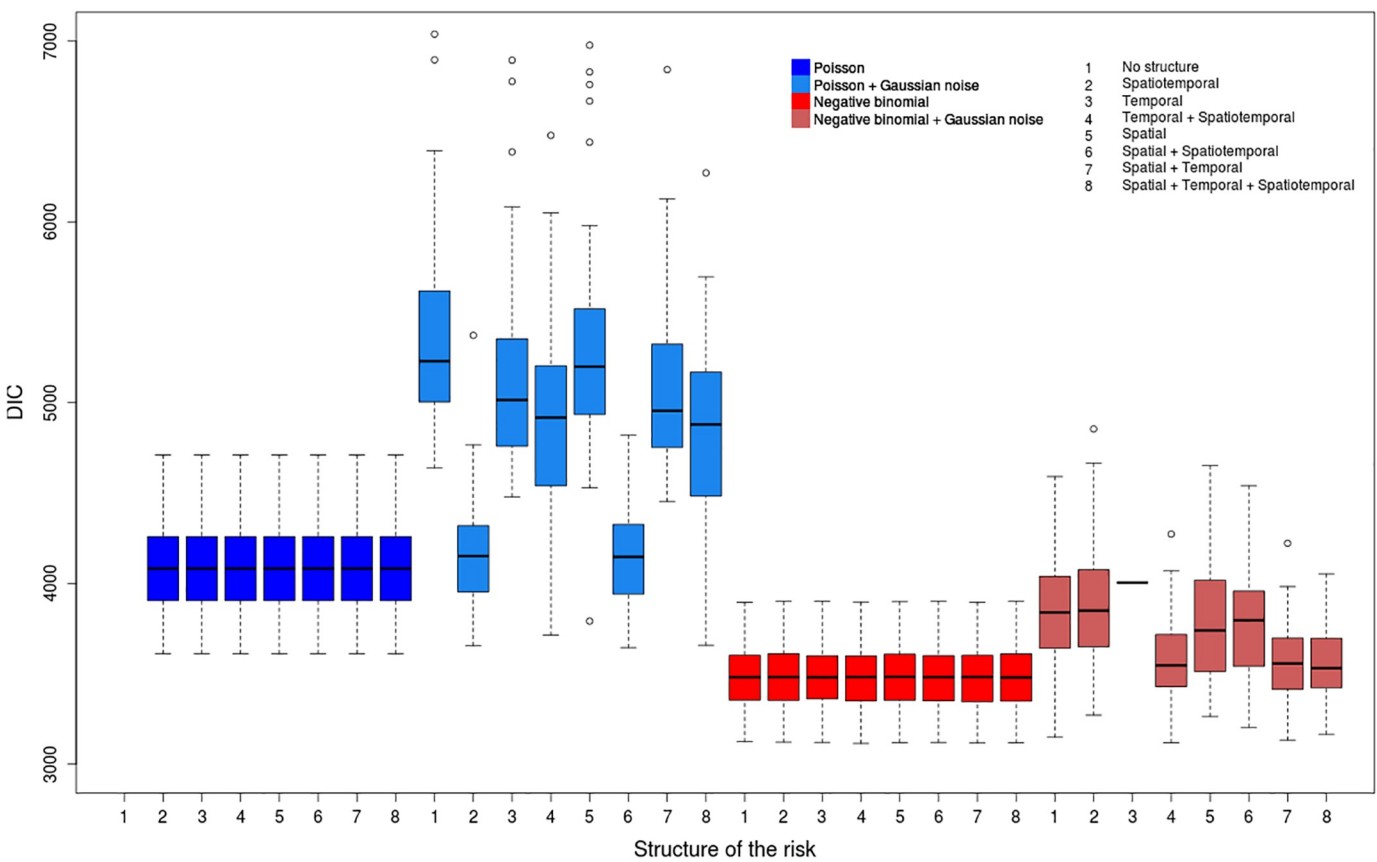

**Fig 5. Distribution of DIC values for the scenario (COSR).**

literature. We also notice that in our context, the Gaussian noise term $\epsilon_{ij}$ is not relevant to model overdispersion for both the Poisson and the negative binomial models, even if this term is used in most of the studies dealing with overdispersed data. Moreover, the use of the Gaussian white noise seems redundant when a negative binomial distribution is tested. For the Poisson models, contrary to what one could expect, the Gaussian noise $\epsilon_{ij}$ does not seem interesting to handle the local overdipersion, regardless of the values of the risk.

**Non-overdispersed highly contrasted data.** We notice that the two best models for the scenario (NOLR) use a negative binomial distribution (Table 2). However, 16 models can be considered as relevant (distance to the best model of average DIC values smaller than 10): 9 of them are negative binomial models and the 7 others are Poisson models (without noise). Nevertheless, if we consider all the models and all the datasets, we can notice the negative binomial models generally better fit the outcome of cases when risk values are highly contrasted (scenarios (COLR) and (NOLR)) (Table 3).

Concerning the distributions of the DIC values for each model, we notice that all the models with the Gaussian component $\epsilon_{ij}$ have a very similar ranking and present small differences of the mean and the variances of their DIC values (Fig 6). Despite the fact that the best model includes a Gaussian noise term (Table 2), in general this parameter is not relevant on average. The negative binomial models with the Gaussian noise $\epsilon_{ij}$ provide slightly higher DIC values than the ones without $\epsilon_{ij}$. Besides, the Poisson models with the Gaussian noise component $\epsilon_{ij}$

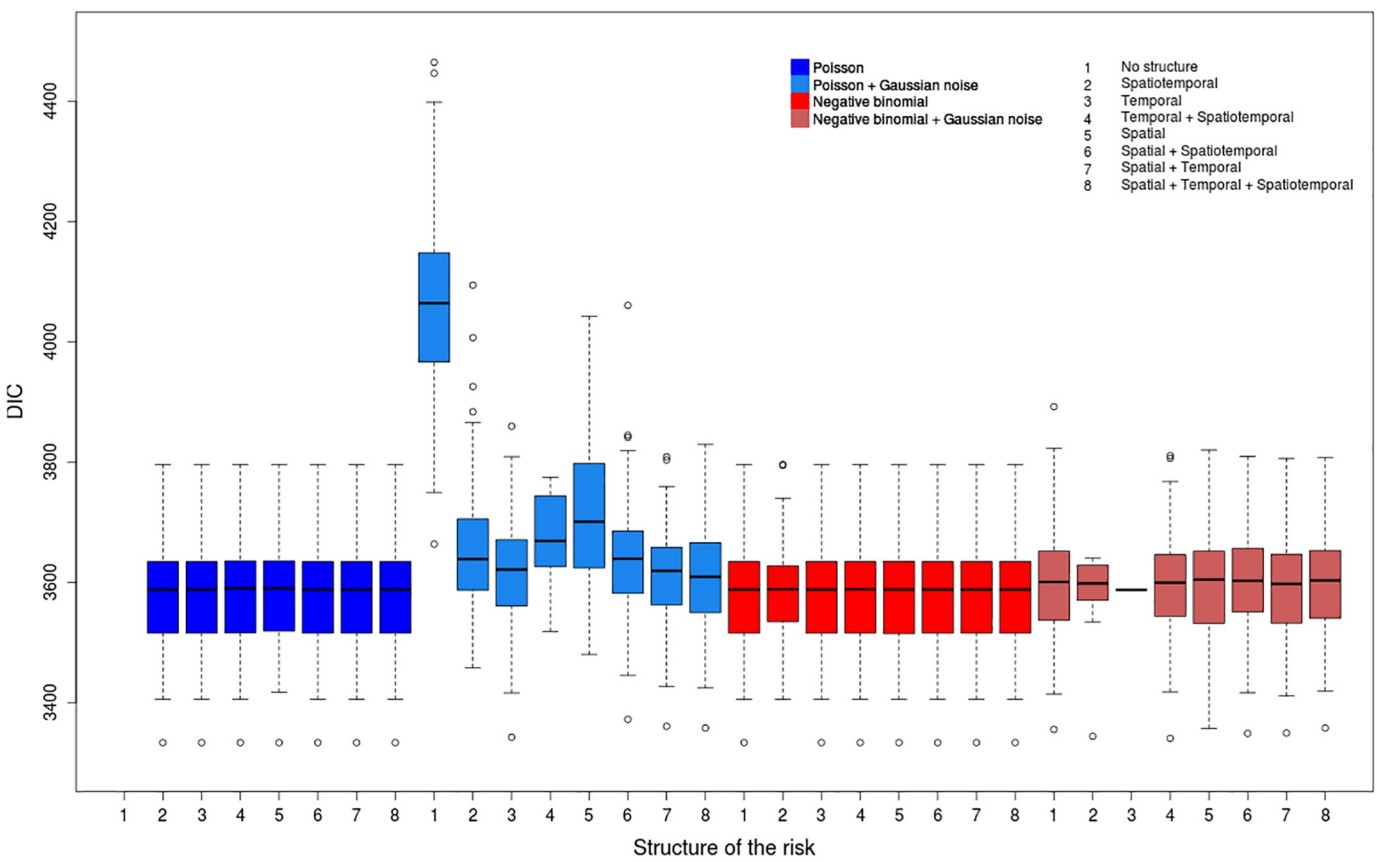

**Fig 6. Distribution of DIC values for the scenario (NOLR).**

have the highest DIC values and show the highest variance (despite their wide use to model overdispersed data in the literature).

The data generated with highly contrasted risk values are overdispersed: the variance of the number of cases by hexagon and by year being much higher than the mean. The heterogeneity resulting from this overdispersion is strongly structured; the Gaussian white noise $\epsilon_{ij}$ is irrelevant to model this variability. In this context, the Poisson and negative binomial distributions show the same relevance.

**Non-overdispersed weakly contrasted data.** Contrary to the scenarios (COLR) and (COSR), the 7 best models for the scenario (NOSR) are all Poisson models, without the Gaussian noise component $\epsilon_{ij}$ (Table 2). They all have a very similar distribution of their DIC values. It is also the case for the negative binomial models, even if their DIC mean values are higher than for the Poisson models (Fig 7). The Poisson models with a Gaussian white noise term $\epsilon_{ij}$ provide on average better DIC values than the negative binomial ones. However the distribution of DIC values for such models is more dispersed.

The DIC values are more similar for all the models for the scenario (NOSR) than for overdispersed data (scenarios (COLR) and (COSR)). For instance, for (NOSR), the average difference of DIC values between the negative binomial and the Poisson models is + 78; this difference is −813.4 for (COLR). The differences of DIC mean values for the models with and without the Gaussian noise $\epsilon_{ij}$ is even more significant for (COLR) than for (NOSR).

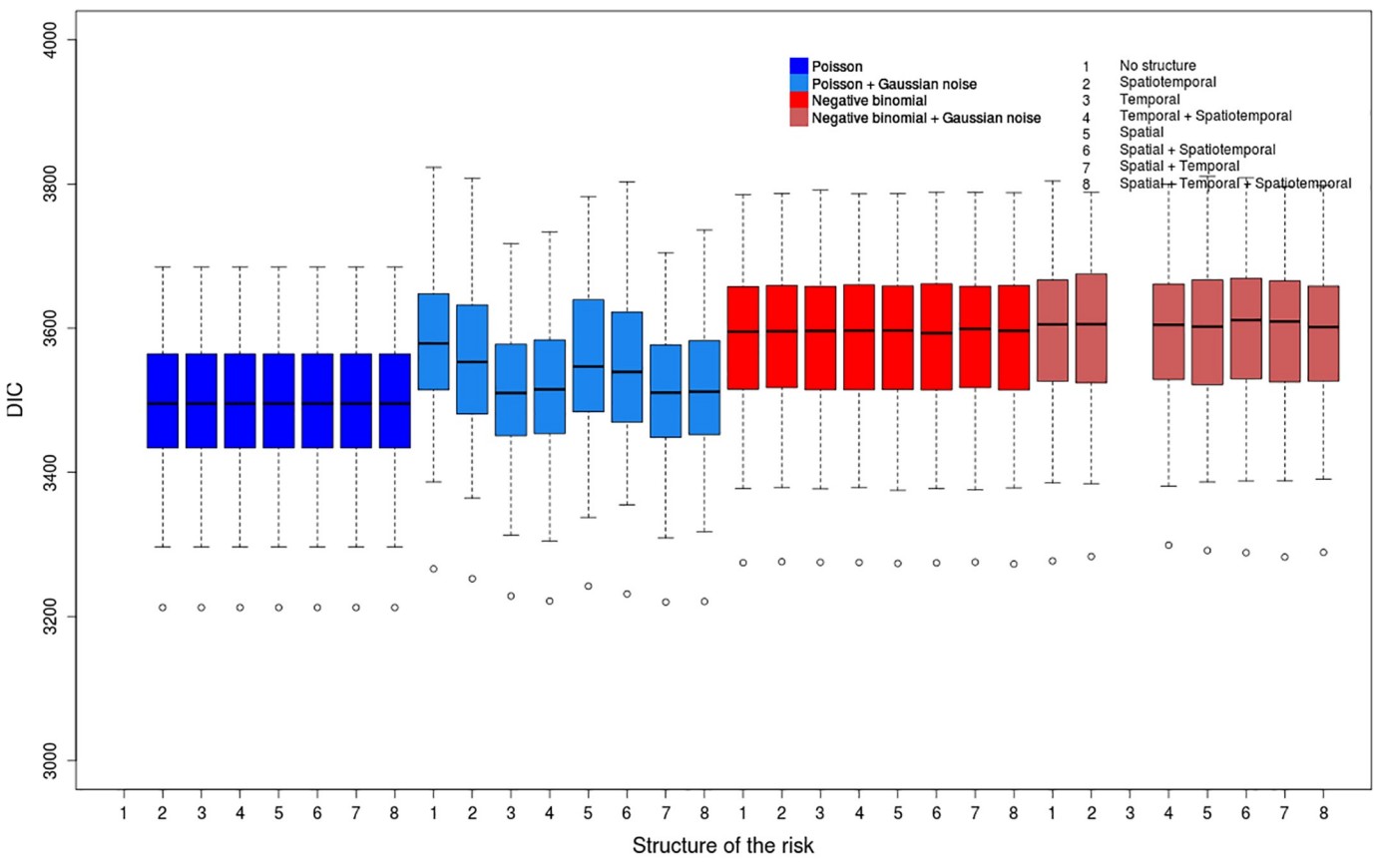

**Fig 7. Distribution of DIC values for the scenario (NOSR).**

As expected, the Poisson distribution appears as the most relevant to model non-overdispersed and weakly contrasted data (NOSR) (Table 3). In this context, the Gaussian noise component is not an interesting approach to model the unstructured heterogeneity, even if its inclusion does not penalize the models a lot.

**Synthesis.** In conclusion, for scenarios (COLR), (COSR) and (NOLR), the negative binomial models without a Gaussian white noise are generally the best ones. It is also the case in general for (NOLR) even if the cases are simulated by a Poisson distribution. It appears that the negative binomial distribution is on average more adapted to both overdispersed data and data with strong differences of levels of risk. At the opposite, the Poisson distribution is particularly well adapted to weakly contrasted and not overdispersed data coming from the scenario (NOSR). For all the scenarios, the Gaussian white noise is generally irrelevant. Thus it appears that the binomial negative distribution is well adapted to handle the overdispersion of data, whether the high heterogeneity is due to local overdispersion (frequent occurrence of high values, high frequence of null values) or to a strong (spatial and temporal) structuration of the risk.

## Handling strong spatiotemporal correlations

Beyond the local overdispersion, the outcome of secondary cases can imply strong spatial and temporal correlations. We aim to test if CAR processes are relevant to handle these dependencies and to determine which CAR components are the most suitable.

**Table 2. Best models according to the DIC.**

| Scenario of simulation | Ranking | Model | DIC | |
|---|---|---|---|---|
| | | | Value | Diff. from the best |
| (COLR) | 1 | nebinnopar_S_x_ST_xxxx | 3561.6 | 0 |
| | 2 | nebinnopar_S_T_xx_xxxx | 3562.8 | +1.2 |
| | 3 | nebinnopar_S_x_xx_xxxx | 3563.0 | +1.4 |
| | 4 | nebinnopar_x_T_xx_xxxx | 3563.3 | +1.7 |
| | 5 | nebinnopar_x_T_ST_xxxx | 3563.3 | +1.7 |
| | 6 | nebinnopar_x_x_ST_xxxx | 3564.7 | +3.1 |
| | 7 | nebinnopar_S_T_ST_xxxx | 3566.7 | +5.1 |
| | 8 | nebinnopar_x_x_xx_xxxx | 3569.1 | +7.5 |
| (COSR) | 1 | nebinnopar_S_T_xx_xxxx | 3486.4 | 0 |
| | 2 | nebinnopar_S_T_ST_xxxx | 3486.6 | +0.2 |
| | 3 | nebinnopar_x_T_ST_xxxx | 3486.6 | +0.2 |
| | 4 | nebinnopar_x_x_ST_xxxx | 3486.7 | +0.3 |
| | 5 | nebinnopar_S_x_ST_xxxx | 3486.7 | +0.3 |
| | 6 | nebinnopar_x_x_xx_xxxx | 3486.8 | +0.4 |
| | 7 | nebinnopar_S_x_xx_xxxx | 3487.2 | +0.8 |
| | 8 | nebinnopar_x_T_xx_xxxx | 3488.5 | +2.1 |
| (NOLR) | 1 | nebinnopar_x_x_ST_gaus | 3573.3 | 0 |
| | 2 | nebinnopar_S_x_xx_xxxx | 3577.9 | +4.6 |
| | 3 | poissnopar_x_x_ST_xxxx | 3578.1 | +4.8 |
| | 4 | poissnopar_x_T_xx_xxxx | 3578.1 | +4.8 |
| | 5 | poissnopar_S_x_ST_xxxx | 3578.1 | +4.8 |
| | 6 | poissnopar_S_T_xx_xxxx | 3578.1 | +4.8 |
| | 7 | poissnopar_S_T_ST_xxxx | 3578.1 | +4.8 |
| | 8 | nebinnopar_x_x_xx_xxxx | 3578.1 | +4.8 |
| | 9 | nebinnopar_x_T_xx_xxxx | 3578.1 | +4.8 |
| | 10 | nebinnopar_S_x_ST_xxxx | 3578.1 | +4.8 |
| | 11 | nebinnopar_S_T_xx_xxxx | 3578.1 | +4.8 |
| | 12 | nebinnopar_S_T_ST_xxxx | 3578.1 | +4.8 |
| | 13 | nebinnopar_x_T_ST_xxxx | 3578.3 | +5.0 |
| | 14 | poissnopar_x_T_ST_xxxx | 3579.6 | +6.3 |
| | 15 | poissnopar_S_x_xx_xxxx | 3580.9 | +7.6 |
| | 16 | nebinnopar_x_x_ST_xxxx | 3581.6 | +8.3 |
| (NOSR) | 1 | poissnopar_x_x_ST_xxxx | 3495.2 | 0 |
| | 2 | poissnopar_x_T_xx_xxxx | 3495.2 | 0 |
| | 3 | poissnopar_x_T_ST_xxxx | 3495.2 | 0 |
| | 4 | poissnopar_S_x_xx_xxxx | 3495.2 | 0 |
| | 5 | poissnopar_S_x_ST_xxxx | 3495.2 | 0 |
| | 6 | poissnopar_S_T_xx_xxxx | 3495.2 | 0 |
| | 7 | poissnopar_S_T_ST_xxxx | 3495.2 | 0 |

**Relevance of CAR processes.** The CAR models have been widely used in the literature to handle the structured heterogeneity. We have tested all the combinations of the three possible CAR components $U_{ij}$, $T_{ij}$ and $V_{ij}$. All these terms are relevant on average to model the structure of the risk for each scenario (Table 3). The temporal CAR component $T_{ij}$ for (COSR) is the only exception.

**Table 3. Average DIC values.**

| Scenario | | (COLR) | (COSR) | (NOLR) | (NOSR) |
|---|---|---|---|---|---|
| Distribution | Poisson Negative binomial | 4592.7 | 4516.0 | 3641.1 | 3517.1 |
| | | 3729.3 | 3617.7 | 3585.3 | 3595.1 |
| Spatial CAR process $U_{ij}$ | No | 4217.4 | 4077.5 | 3627.1 | 3558.0 |
| | Yes | 4111.6 | 4028.8 | 3598.4 | 3554.4 |
| Temporal CAR process $T_{ij}$ | No | 4192.8 | 4045.7 | 3632.8 | 3565.7 |
| | Yes | 4129.1 | 4058.6 | 3593.1 | 3546.5 |
| Spatiotemporal CAR process $V_{ij}$ | No | 4331.0 | 4164.4 | 3628.0 | 3557.7 |
| | Yes | 4012.2 | 3947.4 | 2597.6 | 3554.7 |
| Gaussian white noise $\epsilon_{ij}$ | No | 3857.0 | 3769.6 | 3578.6 | 3546.0 |
| | Yes | 4465.0 | 4317.4 | 3643.9 | 3566.2 |

Nevertheless, the impact of the structure of the risk for the value of the DIC does not seem as significant as the distribution of the cases and the inclusion of the Gaussian white noise $\epsilon_{ij}$. In fact, for each of the 4 scenarios, the differences of average DIC values between the models which include a CAR term and those which do not include it are in general much lower than if one compares the results for each distribution, or for the inclusion of the Gaussian noise $\epsilon_{ij}$ (Fig 3). These differences are particularly low for the scenario (COLR) (Fig 2).

The spatial CAR process $U_{ij}$ was used in the context of purely spatial disease mapping studies. The term $U_{ij}$ and the temporal term $T_{ij}$ were then used in the context of spatiotemporal risk mapping. The spatiotemporal CAR component $V_{ij}$, which assesses the influence of the neighboring areas at the previous and the following periods, is not used a lot in the literature. Beyond the relevance of the CAR terms, our analysis shows that considering both the spatial and the temporal dimensions is well adapted to model the structure of the risk. In fact for the four scenarios, the best models all integrate both dimensions: $V_{ij}$, $U_{ij} + T_{ij}$, $U_{ij} + V_{ij}$, $T_{ij} + V_{ij}$ and $U_{ij} + T_{ij} + V_{ij}$ are in general more relevant than $U_{ij}$, $T_{ij}$ or no structure at all.

**Importance of the structure for overdispersed data.** The two best models for the scenario (COLR) and the five best for the scenario (COSR) include both the spatial and the temporal dimensions. This implies that the overdispersion influences not only the local overdispersion, but also the strength of the structure of the risk. In particular the spatial CAR component $U_{ij}$ is particularly relevant to model the risk associated to the scenario (COLR). Besides, the three best models for the scenario (COSR) integrate at least two CAR terms ($U_{ij} + T_{ij}$, $U_{ij} + T_{ij} + V_{ij}$ and $T_{ij} + V_{ij}$), even if the modeled risk is defined as weakly contrasted. Thus it appears that the overdispersion implies the outcome of well structured cases.

Considering a negative binomial model with the Gaussian noise $\epsilon_{ij}$ fails to model all the heterogeneity of the data, and the remaining heterogeneity is structured. In fact, for the scenario (COLR), the best models use the negative binomial distribution and take into account both the spatial and the temporal dimensions. For the scenario (COSR), the most relevant distribution is also the negative binomial and the best structures of the risk include two or three CAR terms. Conversely, the spatiotemporal structure does not handle all the heterogeneity of the risk, thus the negative binomial distribution is needed. For (COLR) and (COSR), the models which consider the Poisson distribution and the Gaussian noise $\epsilon_{ij}$ show very high DIC values, but the only ones which are acceptable integrate the spatiotemporal CAR process $V_{ij}$. This term seems necessary to offset the variability involved by the Gaussian noise. For both the scenarios (COLR) and (COSR), if we focus on the models which include the negative binomial distribution and the Gaussian noise term $\epsilon_{ij}$, the three best of them have at least two CAR components among $U_{ij}$, $T_{ij}$ and $V_{ij}$.

**Relevance of the spatiotemporal CAR process $V_{ij}$ for highly structured and/or overdispersed data.** Contrary to the spatial and temporal CAR components $U_{ij}$ and $T_{ij}$, the spatio-temporal one $V_{ij}$ is not widely used in the context of disease mapping. However in the framework of our analysis, this term appears as relevant to describe the structure of the risk, sometimes in addition to the CAR components $U_{ij}$ and/or $T_{ij}$. For all the scenarios, including the spatiotemporal CAR term $V_{ij}$ is more relevant than not considering it. Nevertheless, the gain is not really significant in the case of the scenario (NOSR), contrary to the scenarios (COLR), (COSR) and (NOLR), associated to local overdispersion and/or highly contrasted risk values. The best models to describe datasets for which the risk is highly contrasted (scenarios (COLR) and (NOLR)) include the spatiotemporal CAR component $V_{ij}$. For the scenario (COSR), four of the five best models also include this term.

The spatiotemporal CAR component $V_{ij}$ appears as particulary relevant when a Gaussian white noise $\epsilon_{ij}$ is included. The only model considered as the most relevant and which contains the Gaussian noise term $\epsilon_{ij}$ is `nebinnopa_x_x_ST_gaus`, which includes the spatiotemporal CAR process $V_{ij}$. For scenarios (COLR), (COSR) and (NOLR), the best Poisson models with the Gaussian noise term $\epsilon_{ij}$ also integrate the spatiotemporal CAR process $V_{ij}$. The same remark remains true for the negative binomial models with the Gaussian noise component $\epsilon_{ij}$.

## Synthesis for simulated data study

The analysis of the simulated data shows that the negative binomial models are more relevant than the Poisson models to handle locally overdispersed and/or strongly structured data. The Poisson distribution better suits non-overdispersed and weakly structured data. Besides, using negative binomial distribution appears as much more relevant than Poisson models with a Gaussian white noise, contrary to preceding literature. Beyond this local overdispersion, the heterogeneity handled at the second level remains well structured.

The CAR processes, which are used in a wide range of disease mapping studies, appear as a suitable way to take into account this structuration of the data. Overdispersed data, in particular, need the use of several CAR components to model the structure of the risk. Moreover, the spatiotemporal CAR process, whose use is not so frequent, is very relevant in our context and can balance poor modeling choices such as the Gaussian white noise.

The CAR processes and the negative binomial distribution are well adapted to model respectively the structured and the unstructured heterogeneity of the risk. They provide complementary information, thus both are needed to describe the risk.

Lastly the weight coefficients fail to quantify the contribution of each term.

## Application to bovine tuberculosis data

We applied disease mapping to bovine tuberculosis data, both to determine the most relevant models in this context and to test how congruent is the estimated risk with the knowledge about this pathology. We remark that 12 among the 14 the Poisson models with weight parameters ($a$, $b$, $c$) failed to converge. Besides, the two models (`poissparam_S_T_xx_xxxx` and `poissparam_S_T_xx_gaus`) which succeeded in converging showed some estimation errors as their computed DIC value is $+\infty$. It also clearly appears that the DIC is more dispersed in the real data case than in the simulations (S2 Table); indeed the differences between the top-ranked models are higher than in the simulated data case ($\approx 10$ DIC points between the 1st and the 2nd, the 2nd and the 3rd, the 3rd and the 4th, etc.).

The top-ranked model is `nebinparam_S_T_ST_xxxx`. It shows the relevance of the binomial negative distribution compared to the Poisson model with the Gaussian white noise $\epsilon_{ij}$. We can also see that the three CAR components $U_{ij}$, $T_{ij}$ and $V_{ij}$ suit the structure of the risk

**Table 4. Average DIC values.**

| Bovine tuberculosis cases | | |
|---|---|---|
| Distribution | Poisson Negative binomiale | 3128.7 |
| | | 3075.7 |
| Weight parameters $a$, $b$, $c$ | No | 3131.8 |
| | Yes | 3008.1 |
| Spatial CAR process $U_{ij}$ | No | 3108.4 |
| | Yes | 3080.1 |
| Temporal CAR process $T_{ij}$ | No | 3338.7 |
| | Yes | 2878.7 |
| Spatiotemporal CAR process $V_{ij}$ | No | 3117.6 |
| | Yes | 3072.1 |
| Gaussian white noise $\epsilon_{ij}$ | No | 3075.9 |
| | Yes | 3110.0 |

(Table 4). Besides, to address these real data, weight parameters seem suitable to quantify the contribution of each CAR process in the global heterogeneity of the risk when the estimation succeeded. For this best-ranked model, we have $a = 0.84$, $b = 1.83$ and $c = 1.69$, thus

$$\ln(R_{ij}) = 0.84U_{ij} + 1.83T_{ij} + 1.69V_{ij}.$$

Thus it appears that bovine tuberculosis relative risk is highly temporally correlated ($b > a$), congruently with the repartition of the cases (Fig 1). However, all the three terms have a significant influence. Moreover the seven top-ranked models include weight parameters, and the fourteen top-ranked models are negative binomial models, thus our methodological choices (negative binomial with CAR components) appear as relevant to model these infectious disease cases.

Fig 8 shows the estimated bovine tuberculosis risk for the best model (`nebinparam_S_T_ST_xxxx`). We remark that its repartition is quite scattered, so the interpretation is quite difficult for many regions; however, a few conclusions can be drawn. Indeed the risk is particularly high in the Southwest and in Camargue. Besides the risk seems to decrease over the time in the regions which are not classically concerned by bovine tuberculosis. Nevertheless the estimation seems very sensitive to the randomness and overfits the data; indeed the outcome of a case has a strong influence on the risk estimated in the neighborhood for all the periods. The values appear as temporally autocorrelated ($b = 1.83$), thus the outcome of a case impacts the risk values for all the study period, but they are not spatially smoothed enough ($a = 0.84$) and thus not easily interpretable.

The disease map corresponding to the best model (according to the DIC) for the simulated data (COLR) `nebinnopa_S_x_ST_xxxx` seems more relevant to describe the evolution of bovine tuberculosis in France between 2001 and 2010 (Fig 9). We can especially notice that this model is the best ranked one (according to the DIC) which do not include the temporal CAR process. The regions known as being at risk (Dordogne, Côte-d'Or, Camargue, Southwest) are particularly highlighted (S1 Fig). The Center, the North, the Northwest and the Northeast seem free of bovine tuberculosis. We clearly remark that over the time the high risk concentrates itself around a few well-known hotspots. The Southeast shows a high level of risk, but it can be explained by its very low number of cattle farmings that artificially increases the high risk around. A similar problem appears in the neighborhood of Paris which has quite high risk values at the beginning of the study period. In this case too, cattle farmings are rare,

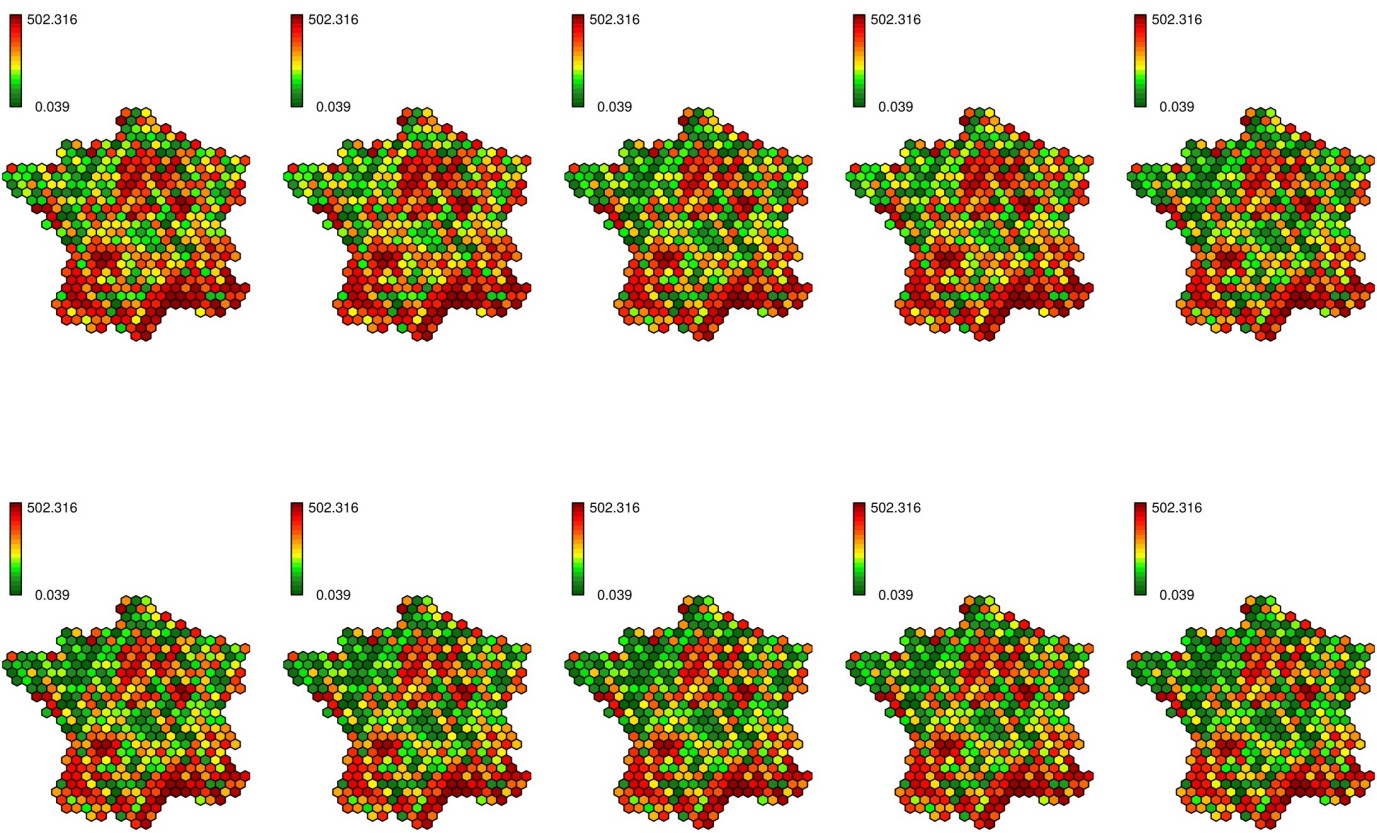

**Fig 8. Estimated risk for the bovine tuberculosis (model `nebinparam_S_T_ST_xxxx`).**

thus the risk estimation is more complicated. Some hexagons show high levels of risk because of the outcome of some isolated cases, but they do not have an influence on their spatial and temporal neighbors. Thus this map shows a relevant (congruent with the literature) and interpretable distribution of the bovine tuberculosis risk.

With the bovine tuberculosis data, it clearly appears that all the models with the temporal CAR component $T_{ij}$ are better ranked than those which do not include this term. However all the maps resulting from these models are hardly readable (Figs 8 and 9, S4 and S5 Figs); the risk is not spatially smooth enough to draw epidemiological conclusions. This may be related to the referenced fact [43, 45] that the DIC could favor overfitted models despite the penalization with the effective number of parameters.

The spatial and spatiotemporal CAR components $U_{ij}$ and $V_{ij}$ provide more interesting maps for epidemiologists. The associated maps are more interpretable as neighboring regions generally show similar risk values.

Models with CAR components as $U_{ij}$, $T_{ij}$ or $V_{ij}$ have better DIC mean values, especially those with $T_{ij}$ (Table 4). In fact, except for two models showing convergence defects, any model including the temporal CAR $T_{ij}$ component outperforms the other models. Thus, the differences of the DIC values between models with and without $T_{ij}$ are particularly high. As expected (according to the ranking of the models for the bovine tuberculosis, see S2 Table), negative binomial models have on average better DIC values than the other ones. Besides, models with weight parameters ($a$, $b$, $c$) and a Gaussian noise $\epsilon_{ij}$ also have better DIC values.

The analysis of bovine tuberculosis data provides results quite similar with those previously obtained for simulated data, in particular for (COLR), except the influence of the temporal

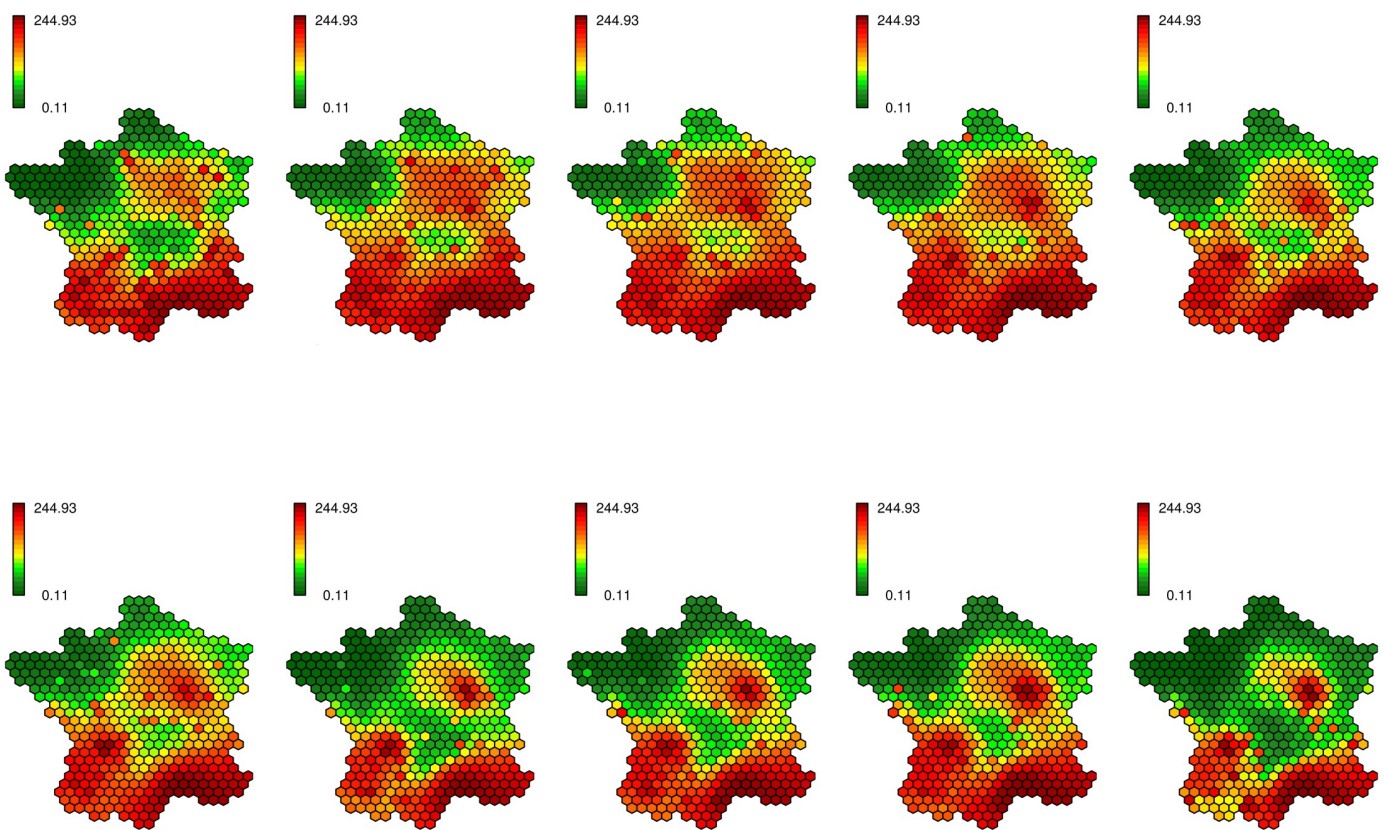

**Fig 9. Estimated risk for the bovine tuberculosis (model `nebinnopar_S_x_ST_xxxx`).**

CAR process. For both real and simulated data, the negative binomial models with different CAR components provide good results according to the DIC. We also remarked that the estimated ranges of the relative risk are more accurate and meaningful with the Negative Binomial than with the Poisson distribution (Figs 8 and 9, S4–S7 Figs) since maximal values are less extreme. Contrary to what is commonly said in the literature, the Gaussian noise term is not relevant in our case, even to model overdispersed data. However, both analyses (simulations and real data) have slight differences. In particular models with weight coefficients are irrelevant for simulated datasets but they are helpful to quantify the contribution of each CAR component to model the bovine tuberculosis data. Thus in the context of our simulations, the three CAR components provide a similar smoothing of the risk.

## Conclusion

This study shows that the overdispersion due to contagion is much better modeled by the negative binomial distribution than by the Gaussian white noise which is commonly used in the literature. In fact the negative binomial models gave better results in the case of overdispersed and/or highly contrasted data. However, it appears that non-overdispersed and weakly contrasted data are better modeled by the Poisson distribution. As other overdispersed distributions, such as the so called "contagious distributions", are known for their goodness-of-fit in the context of infectious diseases, it would be relevant to test them in the framework of disease mapping.

The spatiotemporal framework is relevant to model cases of infectious diseases as, for both real and simulated data (for all the 4 scenarios), the most relevant models integrate both the spatial and the temporal dimensions. On the one hand, the temporal dimension allows to represent the evolution of the spatial distribution of the risk and provides more extensive information about the pathology of interest. On the other hand, the temporal dimension allows the identification of stable areas over the time and thus the confirmation of its status (free or vulnerable) towards the studied phenomenon. In the context of simulated data, it allowed to model the outcome, the movements and the termination of the simulated outbreaks. Moreover, concerning bovine tuberculosis, the estimated risk increases and concentrates around the well known hotspots, congruently with the literature. Besides, the areas at risk were perfectly delineated, and some of the free regions, as for instance the Brittany and the Auvergne, were correctly identified. Thus the temporal dimension, with the temporal and the spatiotemporal CAR processes, provides a real benefit compared to purely spatial studies. In general, the CAR processes satisfactorily take into account the structure of the long range heterogeneity inherent of the data. In particular, the spatial and the spatiotemporal CAR processes provided smoothed risk maps which are epidemiologically relevant and interpretable since the temporal CAR process may result in maps which are harder to interpret. When a single specific pathology is studied, it can be very relevant to integrate populational and/or environmental terms in the structure of the risk. Such cofactors can be considered instead of CAR processes or in addition to them, depending on whether they fully explain or not the structured heterogeneity of the data.

For the analyses of simulated data we performed, the weight coefficients were identified as irrelevant. It may be a consequence of the equilibrium between spatial and temporal correlations in our simulations. However, these weights are highly relevant for the real dataset. Thus it appears that such weights can be useful in case of desequilibrium, for instance when the spatial correlations are stronger than the temporal ones, or conversely. However it would be interesting to test them again in other contexts.

This study shows the relevance of disease mapping to address rare infectious diseases. Most of the simulated hotspots have been identified, even when the risk values are weakly contrasted. The simulated risk was quite difficult to model and the delineation of the estimated risk is not very smooth. It may be due to the very low concentration of the simulated cases which are more scattered than the real datasets. We also remarked that our method overestimates the real relative risk in regions which have a very low cattle population, as previously observed in studies based on purely spatial (and non-contagious) data. Moreover, our methodology provided very relevant maps to represent the estimated bovine tuberculosis risk. These maps are very congruent with the literature concerning the repartition of the risk in France and its evolution between 2001 and 2010. In particular, the detection of simultaneous clusters of cases in three regions is very promising and shows the good capabilities of this methodology to analyse contagious data. The resulting maps also correctly describe the evolution of the risk and its concentration over time around very small areas, congruently with the known contamination of wildlife in these regions. Thus disease mapping appears as a very relevant way to investigate infectious diseases whose incidence is low.

In our study, the model selection criterion is a crucial point as we compared 60 models for each dataset. The DIC (Deviance Information Criterion) is systematically used in the context of spatiotemporal disease mapping and largely used in the purely spatial context. For the simulated data, it provided an interesting ranking of the different models. The best models, according to the DIC, appeared as relevant, because the obtained risk was congruent with the distribution of the simulated risk and the resulting maps were interpretable. Thus this indicator provided interesting evidences concerning the structure and the repartition of the risk.

However, in the bovine tuberculosis context, this criterion seemed to choose overfitted models, for which the estimated risk appeared as too temporally structured, to the detriment of the spatial smoothing. It resulted in maps difficult to interpret, while other non-selected models provided smoother maps which were much more readable for epidemiologists. Thus it seems relevant to study the behavior of the DIC in different situations and eventually propose relevant combinations of criteria or new criteria which take into account the smoothing of the estimated risk. Most of the DIC alternatives try to propose a relevant penalization of the deviance. In contrast, we plan to explore in a forthcoming research how to combine an adequacy indicator such as DIC with smoothing indexes, in order to also take into account the practitioners' need of interpretability. This could be a way to improve the model comparison in the context of risk mapping.

## Supporting information

**S1 Fig. Repartition of cattle farms.**
(TIFF)

**S2 Fig. Map of the 3 outbreaks for simulation.**
(TIF)

**S3 Fig. Map of simulated cases (COLR).**
(TIF)

**S4 Fig. Estimated risk for the bovine tuberculosis (model `nebinnopar_S_T_ST_xxxx`).**
(TIFF)

**S5 Fig. Estimated risk for the bovine tuberculosis (model `nebinparam_S_x_ST_xxxx`).**
(TIFF)

**S6 Fig. Estimated risk for the bovine tuberculosis (model `poissnopar_S_T_ST_xxxx`).**
(TIFF)

**S7 Fig. Estimated risk for the bovine tuberculosis (model `poissnopar_S_x_ST_xxxx`).**
(TIFF)

**S1 Table. Names and characteristics of the considered models.**
(PDF)

**S2 Table. Ranking of the models for the bovine tuberculosis data.**
(PDF)

**S1 Code.**
(PDF)

**S1 File.**
(R)

**S2 File.**
(R)

**S3 File.**
(R)

**S4 File.**
(RDATA)

**S5 File.**
(CPP)

**S6 File.**
(TXT)

## Acknowledgments

We are very grateful to the DGAl (the French Directorate for Food) for providing the Bovine Tuberculosis dataset. We thank Ioana Molnar for her technical assistance with certain figures and scripts. We also thank the reviewers for their interest and their relevant comments and questions.

## Author Contributions

**Conceptualization:** Sylvain Coly, Myriam Garrido.

**Formal analysis:** Sylvain Coly.

**Funding acquisition:** Myriam Garrido.

**Investigation:** Sylvain Coly.

**Methodology:** Sylvain Coly, Myriam Garrido, David Abrial.

**Project administration:** Myriam Garrido, Anne-Françoise Yao.

**Resources:** David Abrial.

**Software:** Sylvain Coly, David Abrial.

**Supervision:** Myriam Garrido, David Abrial, Anne-Françoise Yao.

**Validation:** Myriam Garrido, David Abrial, Anne-Françoise Yao.

**Visualization:** Sylvain Coly, David Abrial.

**Writing – original draft:** Sylvain Coly, Myriam Garrido, Anne-Françoise Yao.

**Writing – review & editing:** Myriam Garrido, Anne-Françoise Yao.

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
