## [Decision Letter · Decision Letter 0]

9 Jan 2020

PONE-D-19-25032

Bayesian hierarchical models for disease mapping applied to contagious pathologies

PLOS ONE

Dear Dr. Garrido,

Thank you for submitting your manuscript to PLOS ONE. After careful consideration, we feel that it has merit but does not fully meet PLOS ONE’s publication criteria as it currently stands. Therefore, we invite you to submit a revised version of the manuscript that addresses the points raised during the review process.

In particular, while both reviewers find your work interesting and worthy of publication, they raise certain major points regarding the statistical analysis, the use of the Deviance Information Criterion (DIC) for model selection and the description of the three outbreak scenarios. 

We would appreciate receiving your revised manuscript by Feb 23 2020 11:59PM. To enhance the reproducibility of your results, we recommend that if applicable you deposit your laboratory protocols in protocols.io, where a protocol can be assigned its own identifier (DOI) such that it can be cited independently in the future. For instructions see: http://journals.plos.org/plosone/s/submission-guidelines#loc-laboratory-protocols

We look forward to receiving your revised manuscript.

Kind regards,

Constantinos I. Siettos, Ph.D.

Academic Editor

PLOS ONE

Additional Editor Comments:

Please make sure that you provide all the necessary information for the reproducibility of the results (codes used to produce the simulations and run the analysis as well as the real data per-se). 

Journal Requirements:

2. Please outline in the Methods section how the data was collected, in enough detail for another researcher to reproduce the findings. For instance, please outline the relevant variables for which data was collected, and the general structure of the dataset, in enough detail for another researcher to request the same dataset. Please also amend your Data availability statement to provide the relevant contact details for where other researchers may apply to access the data. Please also in your Data availability statement clarify whether another researcher can obtain the same dataset, both from the third party and by creating the same simulated datasets.

3. Please note that PLOS ONE has specific guidelines on software sharing (http://journals.plos.org/plosone/s/materials-and-software-sharing#loc-sharing-software) for manuscripts whose main purpose is the description of a new software or software package. In this case, new software must conform to the Open Source Definition (https://opensource.org/docs/osd) and be deposited in an open software archive. Please see http://journals.plos.org/plosone/s/materials-and-software-sharing#loc-depositing-software for more information on depositing your software.

5. We note you have included a table to which you do not refer in the text of your manuscript. Please ensure that you refer to Table 5 in your text; if accepted, production will need this reference to link the reader to the Table.

6. We note that Figures #1,3, 8-11 in your submission contain map images which may be copyrighted. All PLOS content is published under the Creative Commons Attribution License (CC BY 4.0), which means that the manuscript, images, and Supporting Information files will be freely available online, and any third party is permitted to access, download, copy, distribute, and use these materials in any way, even commercially, with proper attribution. For these reasons, we cannot publish previously copyrighted maps or satellite images created using proprietary data, such as Google software (Google Maps, Street View, and Earth). For more information, see our copyright guidelines: http://journals.plos.org/plosone/s/licenses-and-copyright.

a.  You may seek permission from the original copyright holder of Figures #1,3, 8-11 to publish the content specifically under the CC BY 4.0 license.

7. Please upload a copy of Supporting Information Table S1 which you refer to in your text on page 11-12.

Reviewers' comments:

Reviewer's Responses to Questions

**Comments to the Author**

1. Is the manuscript technically sound, and do the data support the conclusions?

Reviewer #1: Yes

Reviewer #2: Yes

2. Has the statistical analysis been performed appropriately and rigorously? 

Reviewer #1: N/A

Reviewer #2: Yes

3. Have the authors made all data underlying the findings in their manuscript fully available?

Reviewer #1: Yes

Reviewer #2: No

4. Is the manuscript presented in an intelligible fashion and written in standard English?

Reviewer #1: Yes

Reviewer #2: Yes

5. Review Comments to the Author

Reviewer #1: DIC was considered for comparing various competing models, which is not theoretically justified since it measures the fit of the model when the parameters are fixed to the posterior expectation. Alternative model selection measures shall be considered such as WAIC.

Examination of goodness of fit (how well the model fits the data) is critical in any statistical analysis to ensure the validity of the statistical analysis, which is not presented in the current manuscript.

Reviewer #2: The manuscript is techinally sound (there are some suggestions that can improve the analysis in the review), the statistical analysis seems to have been performed appropriately and is written in standard English.

Point 3: The authors have not provided the code used to simulate data and run the analysis.

6. PLOS authors have the option to publish the peer review history of their article (what does this mean?). If published, this will include your full peer review and any attached files.

Reviewer #1: No

Reviewer #2: No

---

## [Author Response · Author response to Decision Letter 0]

17 Sep 2020

We thank the reviewers for their interest and their positive advice on the manuscript. Their comments are interesting and raise relevant questions. We have edited our paper to respond to their concerns. In particular, we have clarified and provided some details about the description of the three outbreak scenarios, and also added some references to justify our statements concerning the Deviance Information Criterion (DIC).

More detailed response to the reviwers can be found in the joint Response_to_reviewers file.

---

## [Decision Letter · Decision Letter 1]

20 Nov 2020

Bayesian hierarchical models for disease mapping applied to contagious pathologies

PONE-D-19-25032R1

Dear Dr. Garrido,

We’re pleased to inform you that your manuscript has been judged scientifically suitable for publication and will be formally accepted for publication once it meets all outstanding technical requirements.

Kind regards,

Constantinos Siettos, Ph.D.

Academic Editor

PLOS ONE

Additional Editor Comments (optional):

Reviewers' comments:

Reviewer's Responses to Questions

**Comments to the Author**

1. If the authors have adequately addressed your comments raised in a previous round of review and you feel that this manuscript is now acceptable for publication, you may indicate that here to bypass the “Comments to the Author” section, enter your conflict of interest statement in the “Confidential to Editor” section, and submit your "Accept" recommendation.

Reviewer #2: All comments have been addressed

2. Is the manuscript technically sound, and do the data support the conclusions?

Reviewer #2: Yes

3. Has the statistical analysis been performed appropriately and rigorously? 

Reviewer #2: Yes

4. Have the authors made all data underlying the findings in their manuscript fully available?

Reviewer #2: Yes

5. Is the manuscript presented in an intelligible fashion and written in standard English?

Reviewer #2: Yes

6. Review Comments to the Author

Reviewer #2: All the comments have been adressed and discussed. I find it still a bit pitty that the authors have not employed an MSE type of criterion for comparison. Their justification is fine ("However, as pointed out in the results (page 18) and in the conclusion (page 20) the estimated risk can dramatically increase and reach unrealistic values (sometimes over 200) in sparsely populated areas. In such cases, the MSE values are strongly influenced by these extreme values and fail to provide a relevant ranking of the models."), nevertheless they could have used it on the log scale which would have been less affected by these outliers.

7. PLOS authors have the option to publish the peer review history of their article (what does this mean?). If published, this will include your full peer review and any attached files.

Reviewer #2: **Yes: **Garyfallos Konstantinoudis

---

## [Editor Report · Acceptance letter]

16 Dec 2020

PONE-D-19-25032R1 

Bayesian hierarchical models for disease mapping applied to contagious pathologies. 

Dear Dr. Garrido:

I'm pleased to inform you that your manuscript has been deemed suitable for publication in PLOS ONE. Congratulations! Your manuscript is now with our production department. 

Kind regards, 

on behalf of

Professor Constantinos Siettos 

Academic Editor

PLOS ONE